# HER2 mRNA Levels, Estrogen Receptor Activity and Susceptibility to Trastuzumab in Primary Breast Cancer

**DOI:** 10.3390/cancers14225650

**Published:** 2022-11-17

**Authors:** Tiziana Triulzi, Viola Regondi, Elisabetta Venturelli, Patrizia Gasparini, Cristina Ghirelli, Jessica Groppelli, Martina Di Modica, Francesca Bianchi, Loris De Cecco, Lucia Sfondrini, Elda Tagliabue

**Affiliations:** 1Molecular Targeting Unit, Department of Experimental Oncology, Fondazione IRCCS Istituto Nazionale dei Tumori, 20133 Milan, Italy; 2Nutritional Research and Metabolomics, Department of Experimental Oncology, Fondazione IRCCS Istituto Nazionale dei Tumori, 20133 Milan, Italy; 3Genomic Unit, Department of Experimental Oncology, Fondazione IRCCS Istituto Nazionale dei Tumori, 20133 Milan, Italy; 4Department of Biomedical Science for Health, Università degli Studi di Milano, 20133 Milan, Italy; 5Laboratorio Morfologia Umana Applicata, IRCCS Policlinico San Donato, 20097 Milan, Italy; 6Molecular Mechanisms Unit, Department of Experimental Oncology, Fondazione IRCCS Istituto Nazionale dei Tumori, 20133 Milan, Italy

**Keywords:** breast neoplasms, trastuzumab, mRNA, receptors, estrogen, estradiol

## Abstract

**Simple Summary:**

HER2 mRNA expression is emerging as a powerful predictive biomarker of anti-HER2 drug activity in the treatment of HER2-positive breast cancer patients. Here, we found a positive association between HER2 mRNA levels and disease-free survival in patients treated with adjuvant trastuzumab. Moreover, we found that while HER2 mRNA expression was correlated with the amount of HER2 protein available on the cell membrane in ER-negative tumors, it was correlated with the low ligand-dependent activity of the estrogen receptor (i.e., addiction to HER2) in ER-positive tumors.

**Abstract:**

While the results thus far demonstrate the clinical benefit of trastuzumab in breast cancer (BC), some patients do not respond to this drug. HER2 mRNA, alone or combined with other genes/biomarkers, has been proven to be a powerful predictive marker in several studies. Here, we provide evidence of the association between HER2 mRNA levels and the response to anti-HER2 treatment in HER2-positive BC patients treated with adjuvant trastuzumab and show that this association is independent of estrogen receptor (ER) tumor positivity. While HER2 mRNA expression was significantly correlated with HER2 protein levels in ER-negative tumors, no correlation was found in ER-positive tumors, and HER2 protein expression was not associated with relapse risk. Correlation analyses in the ER-positive subset identified ER activity as the pathway inversely associated with HER2 mRNA. Associations between HER2 levels and oncogene addiction, as well as between HER2 activation and trastuzumab sensitivity, were also observed in vitro in HER2-positive BC cell lines. In ER-positive but not ER-negative BC cells, HER2 transcription was increased by reducing ligand-dependent ER activity or inducing ER degradation. Accordingly, HER2 mRNA levels in patients were found to be inversely correlated with blood levels of estradiol, the natural ligand of ER that induces ER activation. Moreover, low estradiol levels were associated with a lower risk of relapse in HER2-positive BC patients treated with adjuvant trastuzumab. Overall, we found that HER2 mRNA levels, but not protein levels, indicate the HER2 dependency of tumor cells and low estrogen-dependent ER activity in HER2-positive tumors.

## 1. Introduction

*ERBB2* gene, commonly referred to as HER2, is overexpressed or amplified in 15% to 30% of breast cancer (BC) cases, and has been historically associated with poor prognosis and a more aggressive cancer phenotype [1]. The introduction of trastuzumab, the humanized anti-HER2 monoclonal antibody, and the subsequent development of other anti-HER2 agents (i.e., lapatinib, pertuzumab, T-DM1 and T-Dxd) have drastically improved prognosis of HER2-positive BC patients. Nevertheless, not all patients benefit from these agents and eventually experience recurrence. Extensive translational research has been performed to identify predictive markers for patient treatment selection. While HER2 testing by immunohistochemistry and fluorescence in situ hybridization (FISH) and hormone receptor (HR) status are still the gold standard for treatment recommendation, tumor intrinsic features, mainly HER2-addiction [i.e., PAM50, TRAR, estrogen receptor (ER)] and tumor microenvironment features (i.e., tumor infiltrating lymphocytes and immune signatures), have been defined as predictive and/or prognostic biomarkers across multiple clinical trials of anti-HER2 therapies (reviewed in [2,3,4,5]) in accordance with the mechanisms of action of trastuzumab [6].

The role of ER in inducing resistance to HER2 inhibitory reagents has thus far been demonstrated in almost all related clinical trials and real-world studies. Indeed, a higher rate of pathological complete response (pCR) was found in patients with HER2-positive/ER-negative tumors than with ER-positive tumors (reviewed in [7]). Moreover, a recent meta-analysis [8] showed that even the association between pCR and long-term outcomes was strongest in patients with HER2-positive/ER-negative disease receiving trastuzumab. This supported the concept of two distinct diseases. 

HER2 and *ESR1* mRNA levels were found to be associated with the response to anti-HER2 therapies, both as genes belonging to the PAM50 [9], TRAR [10] and 8-gene [11] signatures and as single genes in several neoadjuvant trials, using either trastuzumab, its combination with lapatinib, or pertuzumab independent of the technology used [12,13,14,15,16,17,18]. Notably, HER2 mRNA levels, which are highly heterogeneous among HER2-positive BC specimens [16,19,20,21,22], were found to provide additional response information to PAM50 [17,19], and the combined assessment increased the accuracy of anti-HER2 sensitivity prediction compared to that determined with a single marker [19]. In some of these studies, HER2 and *ESR1* even outperformed clinical parameters (ER, HER2) in predicting pCR [13,16,17]. While in the majority of these studies no subgroup analyses were performed according to tumor ER, Denkert and colleagues found that the pCR rate continuously rose with increased HER2 mRNA levels only in the ER-positive subgroup [12], which is in accordance with the notion that HER2-positive/ER-negative and HER2-positive/ER-positive tumors are different biological entities. At the biological level, several studies carried out in vitro and in animal models, expressing both ER and HER2, demonstrated intricate crosstalk between the two pathways [23].

In this study, we investigated the predictive value of HER2 mRNA levels in HER2-positive BC patients treated with adjuvant trastuzumab according to ER tumor positivity, and analyzed molecular pathways correlated with HER2 mRNA.

## 2. Materials and Methods

### 2.1. Breast Cancer Samples

HER2-positive BC samples of the GHEA cohort were previously described [10]. Briefly, samples were obtained from BC patients treated with adjuvant chemotherapy plus trastuzumab between 2005 and 2009 who did not receive any neoadjuvant treatment (Appendix A, cohort 1). The biospecimens used for research consisted of leftover material of samples collected during standard surgical and medical procedures at Fondazione IRCCS Istituto Nazionale dei Tumori of Milan (INT). Samples were donated by patients to the institutional biobank for research purposes, and aliquots were allocated to this study after approval by the Institutional Review Board and a specific request to the Independent Ethical Committee of INT.

Estrogen levels were measured in serum obtained at the time of surgery from 40 HER2-positive postmenopausal BC patients in the TPM cohort, which consisted of 555 patients with primary nonmetastatic BC who were surgically treated at our institute from 2003 to 2011 [24] and then treated with adjuvant trastuzumab. Appendix A shows the clinicopathological features of these patients (cohort 2). Written informed consent was obtained from all included patients, and the study was approved by the Independent Ethical Committee of INT. All procedures were in accordance with the Helsinki Declaration.

### 2.2. Immunofluorescence Analyses

Immunofluorescence (IF) analyses were performed as described [25]. HER2 was analyzed in 4 μm formalin-fixed, paraffin-embedded (FFPE) slices of the GHEA cohort using rabbit anti-human HER2 antibody (1:50) (Dako, Glostrup, Denmark) after antigen retrieval in 10 mM citrate buffer, pH 6.0, for 6 min at 121 °C, followed by the use of anti-rabbit AlexaFluor 546 secondary antibody (1:1000, Thermo Fisher Scientific, Waltham, MA, USA). Nuclei were stained using 4′,6-diamidino-2-phenylindole (DAPI) Prolong (Thermo Fisher Scientific). Images were acquired on a Leica TCS SP8 X confocal laser scanning microscope (Leica Microsystems GmbH, Wetzlar, Germany). DAPI was excited using a diode laser and detected from 410 to 458 nm, and Alexa Fluor 546 was excited by selecting a 553 nm laser line and detected from 557 nm to 667 nm. Nine images were acquired for each slide using an HC PL APO CS2 40X/1.30 oil-immersion objective, and a pinhole was always set to 1 Airy unit. Data were analyzed using a Leica LAS X rel. 3.1 software (Leica Microsystems GmbH). To avoid fluorescence signal saturation in tumors with the highest HER2 expression, images of all samples were acquired using the microscope setting parameters of the most positive tumor, and protein expression was scored as the mean fluorescence intensity (IF-score) in nine fields of each tumor slide analyzed with a 40x objective.

### 2.3. Estradiol Quantification

Estradiol (E2) in the serum of BC patients was assessed in duplicate using RIA kits (Orion Diagnostica, Espoo, Finland) according to the manufacturer’s instructions. The detection limit of the estradiol kits was 1.36 pg/mL, and the interassay coefficients of variation were 8.5% and 8.9% for mean estradiol titers of 48.3 and 70.5 pg/mL, respectively.

### 2.4. Breast Carcinoma Cell Lines 

In this study, we used the following human HER2-positive BC cell lines which were purchased from the American Type Culture Collection (ATCC): SKBR3, HCC1954, MDAMB361, MDAMB453, BT474 and ZR75.30. The SKBR3 and HCC1954 cell lines were cultured in Roswell Park Memorial Institute (RPMI) 1610 medium (Gibco, Thermo Fisher Scientific); MDAMB361, MDAMB453 and BT474 in Dulbecco’s modified Eagle medium (DMEM) (Gibco) and ZR75.30 in DMEM-F12 (Gibco). All media were supplemented with 10% fetal bovine serum (FBS) (Gibco). All cell lines were grown in a humidified chamber (95% air, 5% CO_2_) at 37 °C and were authenticated (Eurofins Genomics, Ebersberg, Germany). Mycoplasma contamination was assessed every six months using the MycoAlertTM PLUS Mycoplasma Detection kit (Lonza, Basel, Switzerland).

For E2 stimulation, cells were grown for 72 h in RPMI or DMEM without phenol red (Gibco), supplemented with 10% charcoal-stripped FBS (Gibco) and then seeded in the same medium for the experiments as detailed in the results section. Estradiol (10 nM, Sigma Aldrich, St. Louis, MO, USA) or diluent (ethanol) was added, and the cells were analyzed after 24 h.

Tumor cell addiction to HER2 was defined as previously described [26]. Drug titration curves were obtained by treating HER2-positive cells, grown as monolayers in 96-well plates (n = 6 wells/treatment), with lapatinib (from 0.015 to 10 μM, LC Laboratories, Woburn, MA, USA) for 72 h. Control cells were treated with 0.01% DMSO. Cell growth was measured using the sulforhodamine B (SRB) assay: cells were fixed with 10% trichloroacetic acid and then stained with SRB. Absorbance was evaluated at 550 nm, and the growth of treated cells was calculated as a percentage of that in untreated cells.

### 2.5. Fluorescence In Situ Hybridization

The status of HER2/neu was investigated utilizing a commercial probe PathVision HER2/neu DNA probe kit (Vysis, Abbot Molecular, Des Plaines, IL, USA) on metaphase spreads obtained from BC cell lines using standard cytogenetic methodologies [27]. Briefly, slides were pretreated with 2X SSC/0.5% NP40 at 37 °C for 30 min and denatured in Hybrite (Vysis) at 70 °C for 2 min and kept at 37 °C overnight. HER2 gene amplification status was evaluated by applying criteria provided by the manufacturer (Vysis, Abbot Molecular), and we calculated HER2 and centromere for chromosome 17 signals as ratios. FISH hybridization signals were evaluated on at least 60 interphase cells and at least 5 metaphase cells, utilizing an Olympus BX51 microscope coupled to a COHU 4912 charge-coupled device camera (Olympus, Shinjuku City, Tokyo, Japan). The captured images were analyzed using Mac Probe software (PowerGene Olympus, Shinjuku City, Tokyo, Japan).

### 2.6. RNA Extraction and qRT-PCR

RNA was extracted from FFPE tissue slices as described [10]. cDNA was reverse-transcribed in 20 μL volume reaction from 1 μg of total RNA used for microarray analysis [10] with SuperScript III First-Strand Synthesis System and random hexamer primers (Thermo Fisher Scientific). RNA was extracted from BC cell lines using QIAzol (Qiagen, Hilden, Germany) according to the manufacturer’s instructions. cDNA was reverse transcribed from 1 μg of total RNA in a 20 μL volume using the High-Capacity RNA to cDNA kit (Thermo Fisher Scientific). qRT-PCR was performed using TaqMan probe-based assays (HER2: Hs01007077_m1; PGR: Hs00172183_m1) and TaqMan Fast Universal PCR Master Mix on a StepOne Plus^TM^ Real-time PCR system (Thermo Fisher Scientific). The relative abundance of each transcript was calculated by the comparative Ct method, using *GAPDH* (Hs02758991_g1) as a reference gene. 

### 2.7. Flow Cytometry

HER2 protein quantification was performed on live cells by flow cytometry. Briefly, cells were stained with anti-HER2 antibodies produced in our laboratory [MGR2 [28]] and AlexaFluor 488 secondary antibody (1:1000, Thermo Fisher Scientific). Samples were analyzed by gating on live cells after doublet exclusion using the FACSCanto system (BD Bioscience, San Diego, CA, USA) and with the FlowJo software (Tree Star Inc., San Carlos, CA, USA).

### 2.8. Western Blotting and Immunoprecipitation

Protein fractions were solubilized from BC cell lines for 40 min at 0 °C with a lysis buffer containing 50 mM Tris-HCl pH 7.4, 150 mM NaCl, 1% Triton X-100, 2 mM Na-orthovanadate, and a protease inhibitor cocktail (Complete Mini, Roche, Basel, Switzerland). Proteins were resolved by electrophoresis on precast 4–12% bis-tris gels (Thermo Fisher Scientific), transferred to PVDF membranes (Millipore, Burlington, MA, USA) and then incubated with primary antibodies. The following primary mouse monoclonal antibodies were used: HER2 IgG1 (1:300, clone Ab3, Calbiochem, San Diego, CA, USA); mTOR (1:1000, Santa Cruz Biotechnology, Dallas, TX, USA); phospho-mTOR (1:1000, Ser2448, Santa Cruz Biotechnology); p70S6K (1:1000, Santa Cruz Biotechnology); phospho-EGFR (1:200, Tyr1173, Nanotools, München, Germany); and vinculin (1:1000, clone hVIN-1, Sigma-Aldrich). The following primary rabbit polyclonal antibodies were used: phospho-HER2, p-Neu (1:6000, Tyr 1248, Santa Cruz Biotechnology); phospho-p44/42 MAPK (1:1000, Erk1/2, Thr202/Tyr204, Cell Signaling Technology, Danvers, MA, USA); p44/42 MAPK (1:1000, Erk1/2, Cell Signaling Technology); phospho-AKT (1:1000, Ser473, Cell Signaling Technology); AKT (1:1000, Cell Signaling Technology); EGFR (1:800, 1005, Santa Cruz Biotechnology); HER3 (1:200, C-17, Santa Cruz Biotechnology); ER (1:300, HC-20, Santa Cruz Biotechnology), phospho-ER (1:1000, Ser617, Cell Signaling Technology), phospho-p70S6K (1:500, Thr389, Millipore); phospho-4ebp1 (1:1000, Thr 37/46, Cell Signaling Technology), 4ebp1 (1:1000, Cell Signaling Technology), and phospho-HER3 (1:1000, Tyr 1289, Cell Signaling Technology). Filters were then incubated with secondary rabbit or mouse anti-IgG antibodies conjugated with horseradish peroxidase (1:10,000, Amersham, GE Healthcare, Amersham, UK), and proteins were visualized by an enhanced chemiluminescence detection system (Thermo Fisher Scientific). Quantification was performed with Quantity One 4.6.6 software (Bio-Rad, Hercules, CA, USA).

For immunoprecipitation, 4 × 10^6^ BC cells were seeded in 10 mm Petri dishes, cultured for 48 h and lysed as described above. One milligram of the cell extract was precleared by incubation with 80 µL of Sepharose protein A/G (Thermo Fisher Scientific) and 30 µL of normal mouse serum under rocking conditions for 30 min at 4 °C. Then, lysates were incubated under rocking conditions with 1 µg anti-HER2 antibodies produced in our laboratory [MGR2 [28]] for 2 h at 4 °C, followed by incubation with Sepharose protein A/G (Thermo Fisher Scientific) for 3 h while under rocking conditions. Immunoprecipitates were separated, electrophoretically transferred onto nitrocellulose filters, and then probed and revealed as described above.

### 2.9. Antibody-Dependent Cell-Mediated Cytotoxicity (ADCC) Assay

Trastuzumab-mediated ADCC was performed as described [29]. BC cell (BT474, SKBR3, MDAMB453 and MDAMB361) lysis by PBMCs obtained from healthy donors (effector: target ratio 50:1) in the presence of saturating concentrations of trastuzumab (4 μg/mL) was quantified in a 4 h chromium release assay. 

### 2.10. In Silico Analyses

Gene set enrichment analyses were performed using GSEA v4.2.3. Genes were ranked for their correlation with HER2 mRNA levels (ILMN_1728761) or E2 (pg/mL) by Pearson correlation analysis. The analysis involved 1000 gene set permutations, and gene set enrichment was considered significant at FDR < 10%. TCGA reverse phase protein array (RPPA) and gene expression data were downloaded from Broad GDAC Firehose (https://gdac.broadinstitute.org/, accessed on 20 October 2021) and used for correlation analyses. The ER-related score (ERS) score was computed as described [30]. E2 levels were correlated with single genes that were found to be regulated by ligand-dependent (*ESR1*, *XBP1*, *TFF1*) and ligand-independent (*ACP6* and *TNFRSF21)* ER [31].

### 2.11. Statistical Analyses

Statistical analyses were performed with Prism v5 (GraphPad Software, San Diego, CA, USA). Differences between 2 groups were analyzed by unpaired (2-tailed) Student’s *t* test. Correlations between continuous variables were analyzed via Pearson or Spearman correlation. Survival functions were assessed using the Kaplan-Meier estimator, and the log-rank test was used to compare survival distributions. Differences were considered statistically significant at *p* < 0.05.

## 3. Results

### 3.1. Association between HER2 Expression, HER2-Addiction and Patient Prognosis in Human Specimens

Based on the high heterogeneity of HER2 mRNA expression within HER2-positive tumors [16,19,20,21,22], we first validated HER2 expression by qRT-PCR in our cohort of HER2-positive BC patients, treated in adjuvant setting with trastuzumab, for which we have gene expression data that we used to develop the TRAR score ([10], GHEA, n = 36). As shown in Appendix A, HER2 mRNA expression levels among different patients exhibited highly significant correlation in the analyses by microarray and qRT-PCR (r = 0.94, *p* < 0.0001), validating the heterogeneous expression of HER2 mRNA in primary HER2-positive BCs.

As expected, HER2 expression was associated with tumor addiction, as evaluated by the PAM50 [5] or TRAR [10] signature, and significantly higher levels of HER2 mRNA were observed in HER2-enriched (*p* = 0.0006, Figure 1a) and TRAR-low (*p* < 0.0001, Figure 1b) tumors. Accordingly, in the same cohort, high HER2 expression in tumors, defined according to the median HER2 value as cut-off (log2 expression value = 10.37), was not significantly associated with any clinicopathological features (Appendix A) and was related to a significantly better disease-free survival (DFS) probability than low HER2 expression in the entire cohort (Figure 1c) and in the tumor ER expression analysis (Figure 1d,e). HER2 mRNA, evaluated as a continuous variable or using the first tertile as a cutoff to define the HER2 high and low subgroups, maintained its association with DFS in the entire cohort and in the ER subgroups. 

### 3.2. Association between HER2 Expression and HER2 Protein Levels

To examine whether heterogeneity in HER2 mRNA expression levels reflects differences in protein expression, we used quantitative IF analysis to quantify HER2 protein in tumors of the GHEA cohort. We developed an IF-score (see Section 2.2, Figure 2a) that ranged from 35 to 133, showing high heterogeneity in HER2 protein levels in these tumors. The HER2 IF-score did not correlate with HER2 mRNA levels overall (r = 0.20, *p* = 0.2681). Separate analysis according to ER expression showed a significant correlation between HER2 mRNA and protein levels only in ER-negative tumors (r = 0.47, *p* = 0.0584, Figure 2b,c). Accordingly, the HER2 IF-score was significantly associated with DFS only in patients with ER-negative tumors, and high HER2 protein expression was associated with better prognosis (4+–5+ vs. 3+, HR = 0.037, 95% CI = 0.003–0.41, Figure 2d,e). These data suggest that in ER-positive tumors, ER activation could take part in the modulation of HER2 mRNA and/or protein levels.

Notably, a significant strong correlation between HER2 mRNA and HER2 protein levels, as evaluated by IHC, was found in ER-negative BCs but not in ER-positive BCs of a consecutive cohort of BC patients treated in our institute (Appendix A), suggesting that ER-HER2 crosstalk also impacts HER2 mRNA levels in HER2-low BCs. HER2 mRNA correlation with HER2 protein levels, only in ER-negative and not in ER-positive tumors, suggests that ER activation could take part in the modulation of HER2 mRNA and/or protein levels.

### 3.3. Dissection of the Tumor Molecular Features Associated with HER2 Levels

Based on the predictive value of HER2 mRNA levels and on basis of the above-described differences in ER-positive and ER-negative tumors, we performed GSEA in the GHEA cohort to define pathways correlated with HER2 mRNA levels in BC according to ER positivity. As shown in Figure 3, in both ER-negative and ER-positive tumors, HER2 mRNA levels were positively correlated with hallmark pathways related to proliferation and oncogene activation (gray arrows, Figure 3a,b). In ER-positive tumors, HER2 was also found to be positively correlated with immune-related pathways (red bars, black arrows, Figure 3b), glycolysis and HER2 downstream signals (mTORC1 signaling, PI3K/AKT/mTOR signaling, red bars, red arrows, Figure 3b). A significant inverse correlation was found between HER2 mRNA levels and the enrichment of pathways related to ER activity (estrogen response early and late, green bars, Figure 3b). Accordingly, an inverse correlation between HER2 mRNA levels and the ERS score that reflects ER activation [30] was found in these cases (Appendix A).

To better define whether ER activity and/or HER2 downstream signals could affect HER2 mRNA levels, and consequently be related to patient response to treatment, we performed a correlation analysis between HER2 mRNA and proteins of the phosphoproteomic dataset in TCGA (Figure 3c). As found in the GHEA cohort, in ER-positive tumors, negative correlations between HER2 and ER and ER-regulated proteins (PGR, BCL2, IGF1R) were observed. While in both ER-positive and ER-negative tumors, HER2 mRNA levels reflected the activation of the oncoprotein (HER2 and pHER2), mTOR (p4ebp1) and MAPKp38 activation were specifically correlated with ER-positive and ER-negative status, respectively. These data support an inverse correlation between HER2 mRNA expression and ER activity.

To gain deeper insights into the possible regulators of HER2 mRNA levels, we first analyzed HER2 mRNA expression levels according to tumor addiction to the HER2 oncogene and ER expression in 6 HER2-amplified BC cell lines (HER2/CEP17 ratio: BT474, 6.8; ZR75.30, 7.9; MDAMB361, 5.3; SKBR3, 7.1; HCC1954, 6.7, MDAMB453, 5.2). As previously described [26], we confirmed BT474, SKBR3 and ZR75.30 as HER2-addicted cells and MDAMB361 and MDAMB453 as non-HER2-addicted cells, whereas HCC1954 showed an intermediate phenotype similar to that of HER2-addicted cells (Figure 4a). Accordingly, the level of HER2 activation (*p* = 0.0107, Figure 4b,d) and the sensitivity of BC cells to trastuzumab cytotoxic activity (*p* = 0.1073, Appendix A) were higher in HER2-addicted cells than in nonaddicted cells.

As observed in BC patients, HER2 mRNA levels were higher in HER2-addicted BC cell lines (Figure 4c) and were significantly correlated with trastuzumab-dependent cytotoxicity in BC cells (r = 0.96 *p* = 0.0385, Appendix A), indicating that HER2 mRNA levels mirror dependence on the oncogene and predict the response to trastuzumab in BC cell lines. The same was true for ER-positive BC cell lines. Indeed, ER-positive BT474 and ZR75.30 cells (Figure 4d), which are responsive to lapatinib (Figure 4a), have higher HER2 mRNA levels than resistant MDAMB361 cells (Figure 4c). Analysis of HER2 downstream signals in BC cells by WB showed that high activation of mTOR pathway components (pmTOR, pp70S6K) could distinguish ER-positive BC from ER-negative BC cells independently of their HER2 addiction status (Figure 4d). In the analysis of ER-positive cells, HER2 mRNA levels were associated with higher activation of HERs (HER1, HER2, HER3) and their downstream factor pAKT (Figure 4d). pER (Ser167) was activated only in BT474 and MDAMB361 cells, which present higher activation of the mTOR pathway (pmTOR, pp70S6K, p4ebp1) than ZR75.30 cells, in which ERK activation is dominant, suggesting that HER2 mRNA levels are highly associated with HER2 oncogene phosphorylation and PI3K-AKT-mTOR signaling axis activation.

### 3.4. Regulation of HER2 Transcription by Ligand-Dependent ER Activity in BC Cell Lines

To evaluate the relevance of ER in HER2 transcription, we evaluated HER2 mRNA levels in ER-positive BC cell lines (BT474, MDAMB361 and ZR75.30) treated or not treated with fulvestrant, a selective ER degrader. In all cells, fulvestrant significantly reduced the expression of *PGR*, the main target of ER, and HER2 transcription was increased in BT474 and MDAMB361 cells but not in ZR75.30 cells (Figure 5a). To define the relevance of ligand-dependent ER activity in HER2 transcription, we cultured cells in complete medium [containing phenol red and estradiol (E2), the main ER ligand] or in dye-free medium with 10% charcoal-stripped FBS, supplemented or not with E2 (white + E2 and white, respectively, Figure 5b–d). Again, all ER-positive cells had a significantly lower expression of *PGR* when grown in the absence of ER ligands (white medium, Figure 5b), and HER2 mRNA expression was significantly higher in BT474 and MDAMB361 cells grown without ER stimulation than in cells grown with E2 or in a complete medium (Figure 5b). ZR75.30 cells with lower basal *ESR1* and *PGR* expression and high basal levels of ERK activation (Figure 5c,d) did not exhibit increased in HER2 mRNA expression when grown in the absence of estradiol, similar to the results for ER-negative SKBR3 cells (Figure 5b). As shown in Figure 5d, in BT474 and MDAMB361 cells grown in white medium in the absence of E2, an increase in pAKT was observed; in ZR75.30 cells, AKT-mTOR signaling was dramatically reduced in the absence of ER stimulation. 

Overall, these data support the regulation of HER2 transcription by ligand-dependent ER activity in ER-positive HER2-positive BC cells in which the AKT-mTOR pathway is activated and independent of ER (no or low activation by E2 treatment), compared to cells in which this pathway is activated by ligand-dependent ER activity (ZR75.30) (Figure 5d).

### 3.5. Regulation of HER2 Transcription by Ligand-Dependent ER Activity in HER2-Positive BC Patients

To further validate whether HER2 mRNA levels could mirror ligand-dependent ER activity in patients, we measured E2 levels in the available plasma of HER2-positive BC patients of the GHEA cohort (n = 10). We found lower E2 levels in patients with HER2-E tumors than in patients with non-HER2-E tumors (5.1 ± 2.3 vs. 11.2 ± 5.8, *p* = 0.0884), supporting our hypothesis that E2 levels impact the molecular and biological features of HER2-positive BCs. Overall, E2 levels were significantly positively correlated with tumor enrichment in the hallmark estrogen response pathway (red bars, Figure 6a) and with the expression of genes demonstrated to be transcriptionally regulated by ER (*BCL2*, *IGF1R*, *GATA3*, *PIK3CA*) and ligand-dependent ER (*ESR1*, *XBP1*, *TFF1*) [31] (Figure 6c). A nonsignificant inverse correlation was found between the levels of E2 and those of the 2 genes described to be modulated by ligand-independent ER (*ACP6* and *TNFRSF21*, Figure 6c). Even in ER-positive tumors, with the limitation of low samples size (n = 5), E2 maintained its positive correlation with estrogen response pathways by GSEA (red bars, Figure 6b) and with the expression of ER-regulated genes, even if with lower significance. Notably, in this subgroup, E2 levels were inversely correlated with HER2 mRNA levels (r = −0.86, *p* = 0.0645, Figure 6c).

In accordance with the inverse correlation between E2 levels and HER2 mRNA levels and with the prognostic value of HER2 mRNA, we found a significant association of low E2 levels, defined according to the median E2 value as cut-off (10.3 pg/mL) with a lower risk of relapse upon trastuzumab treatment in a cohort of 40 patients with HER2-positive BC, treated with adjuvant trastuzumab (Cohort 2, *p* = 0.0150, Figure 6d). No significant association of E2 levels with patients’ clinicopathological characteristics was observed (Appendix A). The association also maintained its prognostic value in the ER-positive subset (*p* = 0.0114, Figure 6e).

## 4. Discussion

HER2 mRNA expression levels were found to be significantly associated with a better response to HER2-targeted drugs in several clinical trials in HER2-positive BC patients treated with neoadjuvant anti-HER2 drugs [12,13,14,15,16,17,18]. In the adjuvant setting, the only trial analyzed, NSABP B-31 showed that high HER2 mRNA levels are associated with trastuzumab benefit in terms of DFS, especially in patients with ER-positive tumors [32]. Even if based on a small sample cohort, our data provided evidence of the predictive value of HER2 mRNA in both ER-positive and ER-negative subgroups of adjuvant trastuzumab-treated patients, supporting the possibility that HER2 predictive value could be relevant mainly in patients with expression of both ER and HER2, especially in an adjuvant treatment setting. This hypothesis is supported by our data on the predictive value of TRAR, in which *HER2* and *ESR1* are core genes only in ER-positive tumors of the NeoSphere trial [33], and the data of the Gepar IV trial, which showed that pCR rate continuously rose with increases in HER2 mRNA levels only in ER-positive tumors [12].

As previously observed [19], we found higher HER2 mRNA expression levels in HER2-E tumors than in non-HER2-E tumors as analyzed according to the PAM50, which includes *HER2* as one of the 50 genes; higher HER2 mRNA expression was also observed in TRAR-low vs. TRAR-high tumors, supporting the ability of HER2 mRNA to reflect tumor addiction to the signals downstream of the HER2 oncogene and, in turn, its ability to predict the response to trastuzumab. The first obvious possible biological explanation for the predictive value of HER2 mRNA is that it could reflect the amount of the HER2 protein available on the BC cell membranes [4]. Our data support this hypothesis in ER-negative BCs, while in ER-positive tumors, it is more likely that HER2 mRNA only reflects tumor addiction to the oncogene. Indeed, we did not find any correlation between mRNA and protein levels in this patient subgroup, while we found a significant positive correlation between HER2 mRNA and pHER2/pEGFR and an activated downstream pathway, and a negative correlation with ER activity in the TCGA dataset. We are aware of the limitations of our IF-score, i.e., it relies on membrane-associated HER2 and is semiquantitative, but the expected positive correlation between mRNA and protein levels and the predictive value of the HER2 score which we found in ER-negative tumors strongly support our results in ER-positive BCs. Even if there was high concordance between array data of HER2 mRNA and immunohistochemistry/CISH data in consecutive BCs, indicating that HER2 mRNA evaluation could be used as an alternative method to define ER and HER2 positivity [12,21], 10% of the immunohistochemistry/CISH HER2-positive samples were negative by microarray readout [12,34], supporting our findings of HER2 mRNA heterogeneity within HER2-positive BCs, especially ER-positive BCs. Moreover, in the FinHer trial, HER2 protein expression, evaluated by the quantitative assay HERmark, was highly heterogeneous and was not significantly associated with the outcome of patients treated with trastuzumab [35].

The lack of correlation between HER2 mRNA and protein levels could depend on HER2 intratumor heterogeneity, which was not analyzed in this study and not evaluated in the IF-score, but was found to be more frequent in ER-positive tumors and negatively associated with pCR in a study of neoadjuvant T-DM1 [36]. In addition, the high predictive value of HER2 mRNA could just depend on the large dynamic range of mRNA levels within each IHC score group, especially in the IHC HER2-3+ subset observed by us and by others [22], allowing a quantitative continuous evaluation of HER2 positivity. However, considering the correlation between HER2 mRNA and HER2 protein we only found in ER-negative BCs, it stands to reason that in ER-positive tumors, HER2 mRNA, which was found to be regulated by several transcription factors in HER2-negative BCs [37], reflects other signals, which are relevant in cancer sensitivity to trastuzumab, independently of the number of receptors on the cell membrane. Indeed, the described relationship between ER expression and the poor response to trastuzumab and the well-described ER activity as a direct regulator of HER2 transcription [23] support the conclusion that HER2 mRNA levels reflect the activation of ER activity.

Our in vitro data were consistent with these results. Indeed, in BC cell lines, in which overall HER2 mRNA was found to reflect tumor addiction to HER2, pHER2 levels, activation of HER downstream signals (mainly AKT) and response to anti-HER2 drugs, we found an increase in HER2 transcription in cells grown in the absence of estrogens or in which ER degradation is induced by fulvestrant. Our data, even if limited by the use of few BC cell lines, added further complexity even in the ER-positive subgroup. Indeed, the increase in HER2 mRNA expression levels upon E2 deprivation was not observed in ZR75.30 ER-positive cells. These cells are characterized by low basal ER and mTOR activity and high activation of the ERK pathway. Indeed, regarding these pathways, ZR75.30 cells are more similar to ER-negative BC than ER-positive BC, which is generally characterized by higher activation of mTOR and p70S6K and high expression of *PGR*. These cells have low levels of ER phosphorylation derived from HER downstream signals, while the presence of ER ligands induces activation of the AKT/mTOR axis. On the other hand, BT474 and MDAMB361 cells have higher activation of the AKT/mTOR axis, which is why ER ligands did not increase the activation levels. Accordingly, in the TCGA dataset, inverse correlations of *HER2* with *ESR1* and *PGR* were mainly observed in PI3K mutated cases (*ESR1*: r = −0.82, *p* = 0.0038, *PGR: r* = −0.78, *p* = 0.0075), which have higher activation of AKT than PI3K wild type cases (*ESR1*: r = −0.47 *p* = 0.082; *PGR*: r = −0.37 *p* = 0.0372), supporting the hypothesis that in ER-positive tumors with high activation of AKT, estrogens suppress HER2 transcription. The E2-dependent HER2 regulation mechanism seems to be independent of the described regulation of HER2 by AKT activation status [38]. This complex scenario could be the endpoint of the crosstalk between ER and HER [23], ER and mTOR [39] and PI3K pathways [40]. The crosstalk of ER and HER2 with mTORC1, which is the main regulator of protein synthesis, supports the involvement of ER in HER2 mRNA translation [41], contributing to the lack of correlation between HER2 mRNA and protein levels in ER-positive BCs.

The relevance of ligand-dependent ER activation in patients was derived from the significant correlation we found between HER2 mRNA and E2 and with genes demonstrated to be regulated by E2 compared to genes regulated by ER in a ligand-independent manner [31]. Accordingly, a low circulating E2 level was found to be significantly associated with adjuvant trastuzumab benefit in postmenopausal patients. The role of E2 in modulating HER2 partially explain the increase in pCR rate upon anti-HER2 treatment observed in older or postmenopausal versus younger or premenopausal patients [42,43].

From a clinical point of view further studies are necessary to understand whether HER2 mRNA levels could be used as predictive markers of trastuzumab benefit both in ER-positive and ER-negative cohorts and to define usable cutoff. It is worth to investigate, in this context, whether HER2 mRNA could be used also to predict activity of new generation anti HER2-drugs (ADC) both in patients with HER2-positive and with HER2-low BCs. Moreover, based on the notion that PI3K/AKT/mTOR activation is one of the players in trastuzumab resistance [44], ER likely participates in the resistance mechanisms through the crosstalk with components of this pathway and the modulation of HER2 mRNA levels. Thus, in ER-positive tumors, the high degree of crosstalk between ER and the PI3K/AKT/mTOR pathway could inhibit the response to mTOR inhibitors, as observed in the BOLERO-1 and BOLERO-3 trials [45,46], supporting the double targeting of ER and mTOR, which has been performed in ER-positive HER2-negative BC patients [47].

## 5. Conclusions

In summary, this study defined the predictive value of HER2 mRNA levels both in ER-positive and ER-negative HER2-positive BCs treated in an adjuvant setting. In ER-negative tumors, HER2 mRNA expression reflects the abundance of HER2 receptors available for drug binding; in ER-positive BCs, we did not find any correlation between HER2 mRNA and protein levels, and HER2 mRNA levels mainly reflect the activity of the crosstalk between ER, HER2 and the AKT/mTOR pathway.

## Figures and Tables

**Figure 1 cancers-14-05650-f001:**
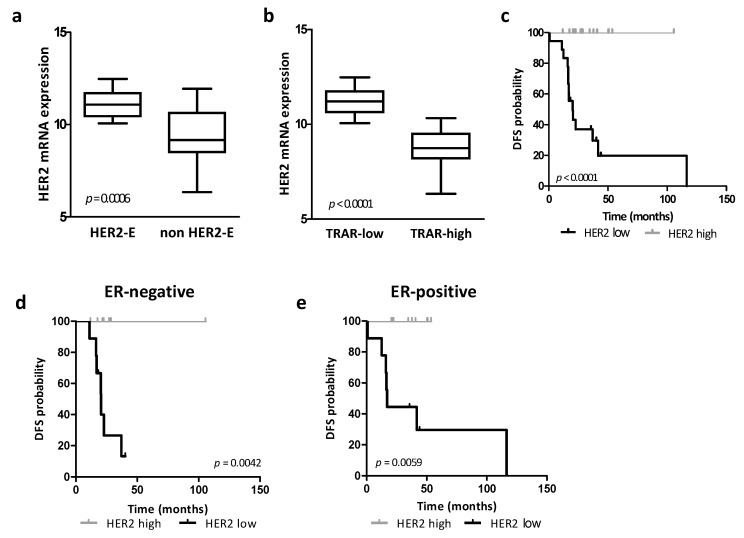
Association between HER2 mRNA expression, HER2-addiction and patient prognosis. (**a**,**b**) Association between PAM50 (**a**) and TRAR (**b**) classification and HER2 mRNA expression in primary HER2-positive BCs of the GHEA cohort. *p* values obtained by Student’s *t* test. (**c**–**e**) Kaplan-Meier analysis of patients in the GHEA cohort classified according to HER2 mRNA levels in the entire cohort (n = 36, **c**), and in ER-negative (n = 18, **d**) and ER-positive (n = 18, **e**) cohorts. The HER2 mRNA median expression value in the entire cohort was used as the cutoff to define the high and low expression groups. *p* values by log-rank test. HER2-E: HER2-enriched.

**Figure 2 cancers-14-05650-f002:**
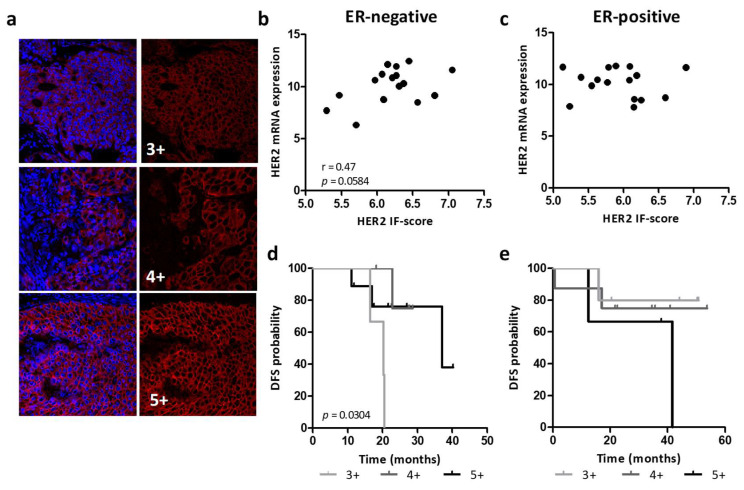
Association between HER2 protein expression and patient prognosis. (**a**) Representative images of HER2 positivity (red) by IF are shown. Images were analyzed using the same parameters. Tumors were scored as 3+ (1st tertile), 4+ (2nd tertile) and 5+ (3rd tertile) according to the mean fluorescence intensities of positive pixels. Nuclei were stained with DAPI (blue). (**b**,**c**) Association between HER2 mRNA expression levels (log2) and protein levels determined according to HER2 IF-score (log2) in ER-negative (**b**) and ER-positive (**c**) subgroups. The Pearson coefficient and relative *p* value are shown. (**d**,**e**) Kaplan-Meier analysis of ER-negative (n = 17, **d**) and ER-positive (n = 16, **e**) patients classified according to the HER2 IF-score. *p* value determined by log-rank test.

**Figure 3 cancers-14-05650-f003:**
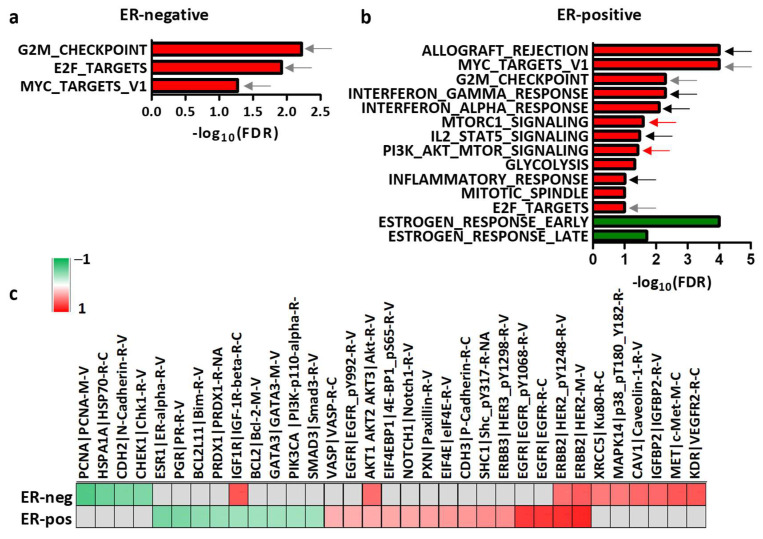
Correlation between HER2 mRNA and tumor molecular characteristics according to ER expression. (**a**,**b**) Bar plot showing HALLMARK pathways significantly (FDR < 10%) and positively (red bars) or negatively (green bars) correlated with HER2 mRNA in ER-negative (**a**) and ER-positive (**b**) HER2+ BCs of the GHEA cohort as evaluated by GSEA. FDR: false discovery rate. Gray arrows: proliferation-related pathways; red arrows: oncogene downstream signaling pathways; black arrows: immune-related pathways. (**c**) Proteins significantly (*p* < 0.05) correlated with HER2 mRNA levels in the TCGA dataset. Pearson’s r is color-co3.4. Association between HER2 Expression and HER2-Addiction in BC Cell Lines.

**Figure 4 cancers-14-05650-f004:**
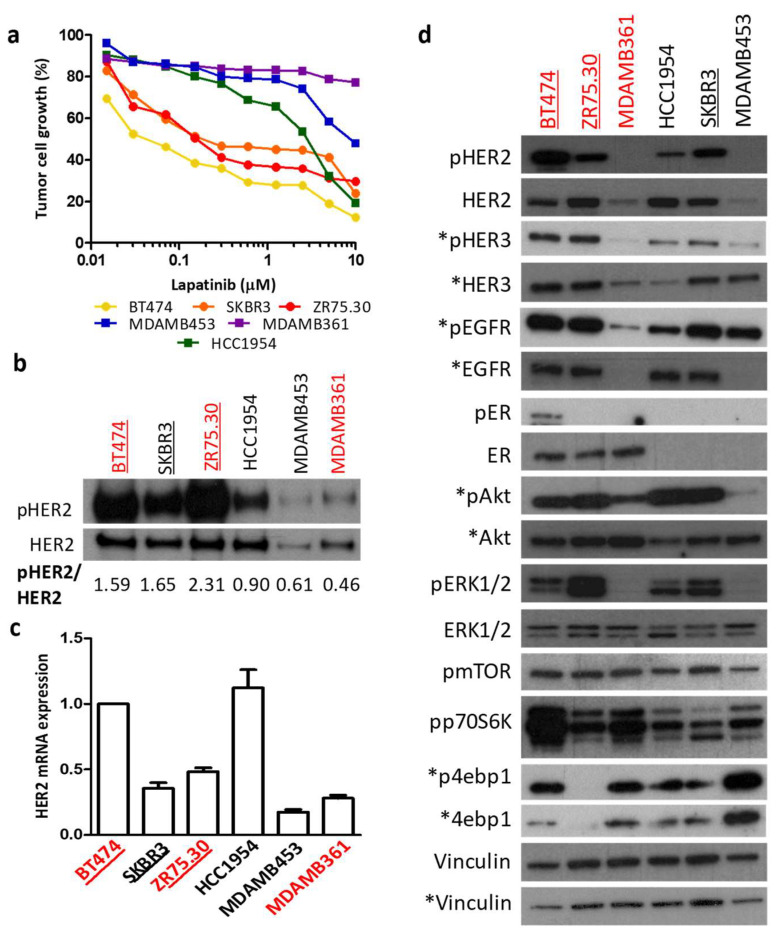
Characterization of HER2-positive BC cell lines. (**a**) Lapatinib dose–response curves for HER2-positive BC cell lines. Results are normalized to those of untreated cells and are representative of two independent experiments. (**b**) Western blot analysis of HER2-positive BC cell extracts after immunoprecipitation with anti-HER2 antibodies. Images are representative of two independent experiments. Ratios were calculated for band quantification by Quantity One. ER-positive and HER2-addicted BC cell lines are shown in red and underlined, respectively. (**c**) HER2 mRNA levels as evaluated by qRT-PCR. Data are the mean ± SD of three independent experiments and are relative to BT474 HER2 expression levels. (**d**) Western blot analysis of HER2-positive BC cell extracts. Vinculin was used as a loading control. Images are representative of two independent experiments. The names of ER-positive and HER2-addicted BC cell lines are shown in red and underlined, respectively. The asterisks indicate molecules analyzed in a separate gel.

**Figure 5 cancers-14-05650-f005:**
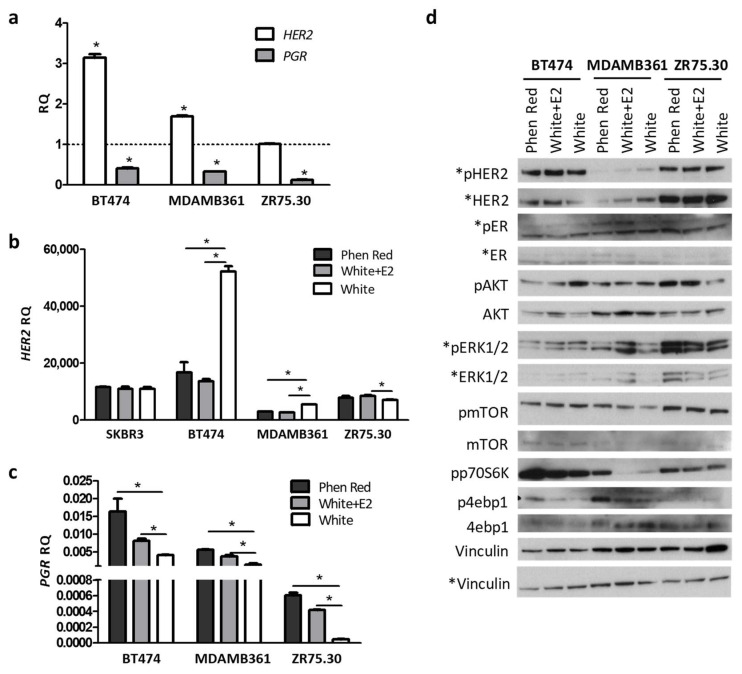
Modulation of ligand-dependent ER activity in ER-positive, HER2-positive BC cell lines. (**a**) HER2 and PGR mRNA levels, as evaluated by qRT-PCR in cells grown in complete medium and treated with 10 nM fulvestrant for 24 h. Data are relative to cells treated with diluent (dotted line), are the mean ± SD of technical replicates and are representative of 3 independent experiments. (**b**) HER2 mRNA levels as evaluated by qRT-PCR in cells grown in complete medium containing 10% FBS (with phenol red) or in white medium with 10% charcoal-stripped FBS, treated with 10 nM estradiol (white + E2) or not (white) for 24 h. Data are the mean ± SD of technical replicates and are representative of 3 independent experiments. (**c**) PGR mRNA expression in cells treated as in (**b**). Data are the mean ± SD of technical replicates and are representative of 3 independent experiments. (**d**) Western blot analysis of cells treated as in b. Representative of two independent experiments. Vinculin was used as a loading control. The asterisks indicate molecules analyzed in a separate gel. * *p* < 0.05 by unpaired Student’s *t* test. RQ: relative quantification.

**Figure 6 cancers-14-05650-f006:**
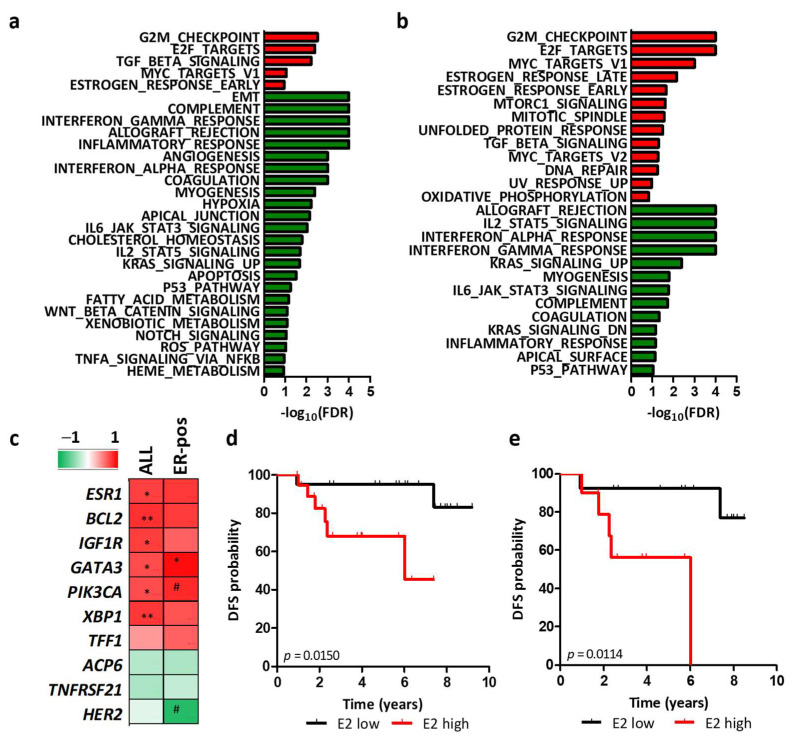
Correlation between E2 levels and tumor characteristics in BC patients. (**a**,**b**) Bar plot showing HALLMARK pathways significantly (FDR < 15%) positively (red bars) or negatively (green bars) correlated with E2 (pg/mL quantified in plasma of patients at surgery) in 10 postmenopausal HER2-positive BC patients of the GHEA cohort (**a**) and in the ER-positive subgroup (n = 5, **b**) as evaluated by GSEA. FDR: false discovery rate. (**c**) Correlation of E2 with the expression of known ER-regulated genes in the same cohort analyzed in a. Pearson’s r is color-coded, and its significance is shown: # *p* < 0.1, * *p* < 0.05, ** *p* < 0.01. (**d**,**e**) Kaplan-Meier analysis of postmenopausal patients with HER2-positive BC treated with adjuvant trastuzumab classified according to circulating E2 levels in the entire (E2 high n = 20, E2 low: n = 20, **d**) and ER-positive (E2 high n = 10, E2 low: n = 13, **e**) cohorts. The E2 median value in the entire cohort was used as a cutoff to define the high- and low-level groups. *p* values by log-rank test.

## Data Availability

All data are present in the paper or the Appendix A. Gene expression profiles of the human BCs that were analyzed in this paper were previously deposited to the Gene Expression Omnibus database, series GSE55348.

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
