# Peer review of "HER2 mRNA Levels, Estrogen Receptor Activity and Susceptibility to Trastuzumab in Primary Breast Cancer"

_cancers, 2022, doi:10.3390/cancers14225650_

Round 1
Reviewer 1 Report
The authors investigated the predictive value of HER2 mRNA levels in HER2-positive BC patients treated with adjuvant trastuzumab according to ER positivity and analyzed molecular pathways correlated with HER2 mRNA. And they found a positive association between HER2 mRNA levels and disease-free survival in patients treated with adjuvant trastuzumab. Moreover, they found that while HER2 mRNA expression was correlated with the amount of HER2 protein in ER-negative tumors, it was correlated with low ligand-dependent activity of the ER in ER-positive tumors.
This article is very interesting and well organized. However, this study was performed with a small number of the patients and caution is needed in generalizing the findings of this study.
Nevertheless, this study can help readers a lot. I ask that authors should revise it based on the below commentary.
(1) Throughout the manuscript, HER2 mRNA and ERBB2 mRNA are mixed and used. Please unify the two terms.
(2) Please correct the keywords to MESH term.
(3) What’s the range of the HER2 mRNA? What’s the definition of the low and high HER2 mRNA? And what’s the evidence? Please describe additional information related to this.
(4) In the ‘Results’ 3.3, where is the green bars in Figure 3b? (301th line) Please check.
Author Response
The authors investigated the predictive value of HER2 mRNA levels in HER2-positive BC patients treated with adjuvant trastuzumab according to ER positivity and analyzed molecular pathways correlated with HER2 mRNA. And they found a positive association between HER2 mRNA levels and disease-free survival in patients treated with adjuvant trastuzumab. Moreover, they found that while HER2 mRNA expression was correlated with the amount of HER2 protein in ER-negative tumors, it was correlated with low ligand-dependent activity of the ER in ER-positive tumors.
This article is very interesting and well organized. However, this study was performed with a small number of the patients and caution is needed in generalizing the findings of this study.
Nevertheless, this study can help readers a lot. I ask that authors should revise it based on the below commentary.
(1) Throughout the manuscript, HER2 mRNA and ERBB2 mRNA are mixed and used. Please unify the two terms.
Response. We thank the reviewer for his/her positive comments and as requested we unified the use of HER2 mRNA throughout the manuscript in accordance with what we used in the title.
(2) Please correct the keywords to MESH term.
Response. The keywords have been changed to MESH terms as suggested. New keywords are: breast neoplasms; trastuzumab; mRNA; receptors, estrogen; estradiol.
(3) What’s the range of the HER2 mRNA? What’s the definition of the low and high HER2 mRNA? And what’s the evidence? Please describe additional information related to this.
Response. The range of HER2 mRNA level in our cohort was (log2expression value: 6.34-12.47). The HER2 mRNA median expression value (log2expression value=10.37) was used as a cutoff to define the high and low expression groups as detailed in Figure1 legend. We used HER2 mRNA as dichotomic variable to show Kaplan-meyer curves. We chose median value as cut-off without any intent to propose this cut-off as usable one. In the few manuscripts in which HER2 mRNA predictive value was evaluated as dichotomic variable in HER2-positive breast cancers, median (Bianchini et al BCR 2017) or tertile (Prat et al, JNCI 2020), or ad hoc cut-off that in TCGA correspond to the 50% percentile (Griguolo et al Cancers 2020) were used. We are aware of the difficulties to choose a ‘correct cut-off’ with such few patients and without validation. That’s the reason why all the other analyses in the manuscript were performed with HER2 as continuous variable.
Based on the request of the reviewer, we performed further analyses to confirm the predictive value of HER2 mRNA in our cohort independently from the cut-off used. HER2 mRNA was found associated with benefit from trastuzumab both as continuous variable (entire cohort, HR=0.46, 95%CI: 0.32-0.66; ER negative, HR=0.57, 95%CI: 0.35-0.91; ER-positive, HR=0.20, 95%CI: 0.06-0.65), and as dichotomic variable, using the first tertile of the entire cohort as cut-off to define low and high HER2 (figure below: all, HER2 low n=12, high n=24; ER-negative HER2 low n=6, high n=12; ER-positive HER2 low n=6, high n=12).
We have now specified in the text that we used median cut-off (we also added the value) to define HER2-high and low subgroups and we have added a sentence in the results indicating that HER2 is predictive both as continuous variable and using the first tertile as cut-off.
(4) In the ‘Results’ 3.3, where is the green bars in Figure 3b? (301th line) Please check.
Response. Green bars are marking HALLMARK pathways significantly (FDR<10%) negatively correlated with ERBB2 (Estrogen response early and late) and red bars the positively correlated ones, as detailed in the figure legends. To avoid misinterpretation, and make the text easier to follow we have now specified it also in the main text.

Reviewer 2 Report
Are there correlations between ERBB2 expression and clinicopathological characteristics of the tumor? The conclusions of the authors are based on the analysis of 40 patients, did I understand correctly? Were all patients treated with trastuzumab? How many courses?
Minor remarks:
1. In the Results section, there should be no references to literary sources.
2. Figure 3, 4, 6 some signatures are not readable.
Author Response
Reviewer 2
Are there correlations between ERBB2 expression and clinicopathological characteristics of the tumor? The conclusions of the authors are based on the analysis of 40 patients, did I understand correctly? Were all patients treated with trastuzumab? How many courses?
Response. Yes, in this study we analyzed two small cohorts of HER2-positive breast cancer patients treated with adjuvant trastuzumab (cohort 1, n=36 in which we analyzed gene expression profile, and cohort 2, n=40, in which we have estradiol quantification. The two cohorts overlapped for 10 samples). We are aware that our conclusions are based only on few cases, thus we clearly stated it in the discussion section as a limitation of the study.
All the patients received adjuvant trastuzumab in their adjuvant treatment schedule. The 1-year trastuzumab treatment, as per treatment guidelines, was completed in 34 (94.4%) patients of cohort 1 and 38 (95%) patients of cohort 2. Discontinuation of the treatment was for reasons other than relapse and are mainly related to toxicities.
HER2 mRNA was not associated with any clinicopathological features available (table below that is the new Table S1 of the manuscript), as shown below for cases splitted by median cutoff. Similar results were obtained for cohort 2 according to E2 levels. T
Table 1. Clinicopathological characteristics of HER2-positive breast carcinoma patients according to HER2 mRNA levels.
Characteristic |
HER2 low n=18 (%) |
HER2 high n=18 (%) |
E2 low n=20 (%) |
E2 high n=20 (%) |
Age Median (interquartile range) |
53 (47-58) |
55 (50-59) |
58 (56-64) |
60 (57-64) |
Size ≤ 2 cm |
10 (56) |
9 (50) |
12 (60) |
15 (75) |
> 2 cm |
8 (44) |
9 (50) |
8 (40) |
5 (25) |
Grade II |
4 (22) |
4 (22) |
6 (30) |
2 (10) |
III |
14 (78) |
14 (78) |
14 (70) |
18 (90) |
Nodeb N0 |
3 (17) |
2 (11) |
9 (45) |
6 (30) |
N+ |
15 (83) |
16 (89) |
11 (55) |
14 (70) |
ERa neg |
9 (50) |
9 (50) |
7 (35) |
10 (50) |
pos |
9 (50) |
9 (50) |
13 (65) |
10 (50) |
PgRa neg |
10 (50) |
8 (60) |
13 (65) |
11 (55) |
pos |
8 (50) |
10 (40) |
7 (35) |
9 (45) |
BMI <20 |
0 |
0 |
1 (1) |
0 |
≥20<25 |
3 (17) |
5 (28) |
15 (75) |
12 (70) |
≥25<30 |
1 (5) |
0 |
4 (20) |
4 (20) |
≥30 na |
0 14 (78) |
0 13 (72) |
0 0 |
2 (10) 2 (10) |
a ER- and PgR-positive, > 10% cell positivity by immunohistochemistry
b Node status positive, at least one positive lymph node by histological examination
Minor remarks:
- In the Results section, there should be no references to literary sources.
Response. We checked all references of the text and we moved those cited for the first time in the result section in introduction or in materials and methods. Then, we maintained citation of some of them in the results section because we think it could help readers in the comprehension of the text. In details new ref 30, first cited in the materials and methods was maintained also in the results to underline that the ERS score reflects ER activation; new ref 26 and 31 moved to materials and methods, were re-cited in the results to give immediately to readers the feedback on the congruence to the literature of our data related to HER2-addiction of our breast cancer cell lines and on details of ER-dependent genes analyzed in our cohort, respectively.
- Figure 3, 4, 6 some signatures are not readable.
Response. Texts of the heatmaps of figure 3-4-5 have been now enlarged to make all signatures readable.